Protein composition analysis of human plasma-derived and recombinant human serum albumin preparations based on 4D label-free proteomics

Ma Li 1
Jiang Peng 1
Wang Zongkui 1
Liu Qing 1
Xu Jun 2
Cheng Lu 2
Sun Pan pan.sun@ibt.pumc.edu.cn sunpan_1@163.com 1
Du Xi shela44@163.com 1
Li Changqing 1
1 Institute of Blood Transfusion, Chinese Academy of Medical Sciences & Peking Union Medical College , Chengdu , China
2 Research and Development Department, Shanghai RAAS Blood Products Co., Ltd , Shanghai , China
Uversky Vladimir
Electronic publication date: 2025 Jun 30
Publication date: 2025
Volume: 13
Electronic Location ID: e19624
Received 2025 Jan 16; Accepted 2025 May 30
Copyright: ©2025 Ma et al.
Copyright year: 2025
Copyright holder: Ma et al.
License: This is an open access article distributed under the terms of the Creative Commons Attribution License, which permits unrestricted use, distribution, reproduction and adaptation in any medium and for any purpose provided that it is properly attributed. For attribution, the original author(s), title, publication source (PeerJ) and either DOI or URL of the article must be cited.
License URL: https://creativecommons.org/licenses/by/4.0/

Keywords: 4D label-free, Proteomics, Human serum albumin, Accompanying proteins

Funding: CAMS Innovation Fund for Medical Sciences 2021-I2M-1-060 Sichuan Science and Technology Program 2022YFS0010 This study was funded by the CAMS Innovation Fund for Medical Sciences (Grant No.2021-I2M-1-060) and Sichuan Science and Technology Program (Grant No. 2022YFS0010). The funders had no role in study design, data collection and analysis, decision to publish, or preparation of the manuscript.

==============================
Background

Clinical therapeutic human serum albumin (HSA) preparations are typically derived from human plasma and contain various accompanying proteins (APs). Previous studies have documented extensively the disparities in post-translation modifications, redox states and antioxidant capacities among HSA preparations from different manufacturers. Most of these studies have focused primarily on albumin, and analyzing APs in HSA preparations and recombinant HSA (rHSA) was often neglected.

Methods

In this study, the APs in human plasma-derived HSA (pHSA) from six Chinese manufacturers and recombinant HSA (rHSA) from yeast and rice were identified and analyzed using a four-dimensional (4D) label-free quantitative proteomic technology.

Results

A total of 456 different APs from the six pHSA preparations were identified, with 96 APs consistently detected in all pHSA samples. 52 APs from yeast-produced rHSA were identified, whereas 152 APs were detected in rice-expressed rHSA. Among the detected APs, haptoglobin, hemopexin and transthyretin were among the top eight APs with the highest relative abundance consistently observed in all pHSA preparations. Moreover, the results revealed that the identified APs in pHSA are primarily involved in endopeptidase inhibitor activity, complement and coagulation cascades, biosynthesis of amino acids and cholesterol metabolism by Gene Ontology (GO), Clusters of Orthologous Groups (COG)/euKaryotic Orthologous Groups (KOG), Kyoto Encyclopedia of Genes and Genomes (KEGG) and protein–protein interactions (PPI) annotation. The ELISA validation results confirmed the presence of haptoglobin, hemopexin, transthyretin and serotransferrin in pHSA but not in rHSA, aligning with the findings from the 4D label-free quantitative proteomic analysis.

Introduction

In the 1940s, during World War II, the US Army surgeon Isidor S. Ravdin successfully pioneered the clinical application of human serum albumin (HSA). HSA products are used to treat various diseases, including shock, trauma, bleeding, acute respiratory distress syndrome, hemodialysis, acute liver failure, chronic liver disease and hypoalbuminemia (Fanali et al., 2012). Therapeutic plasma-derived HSA (pHSA) preparations are typically produced from pooled plasma of health donors by low-temperature ethanol precipitation, which is often referred to as Cohn’s Method (Cohn et al., 1946). This process yields an albumin solution with a yield of approximately 83% and a purity exceeding 96% (Peters, 1995), thus meeting the purity requirements of pHSA outlined by pharmacopeias such as the European Pharmacopoeia and the Chinese Pharmacopoeia (not less than 96%) (European Pharmacopoeia Commission, 2023). Despite the emphasis on purity, the current production process cannot eliminate proteins that share similar physical and chemical properties with albumin. Consequently, several accompanying proteins (APs) are inevitably present in the final pHSA products. Significant variations in the characteristics of commercial HSA preparations have been revealed and arise because of differences in the manufacturing processes and plasma pools used by different manufacturers (Bar-Or et al., 2005). These substantial differences are primarily caused by post-translational modification, redox status and antioxidant properties of HSA preparations produced by different manufacturers (Kleinova et al., 2005; Marie et al., 2013; Miyamura et al., 2016; Nakae et al., 2017; Plantier et al., 2016). However, these studies have primarily focused on albumin, and comprehensive analyses and comparisons of the various protein components in commercial HSA products have not been conducted.

Several studies have identified a number of APs in therapeutic HSA solutions. In a previous study, proteomic analysis of one single commercial albumin preparation successfully identified an additional 141 proteins (Gay et al., 2010). In contrast, in a recent study, only 65 APs were identified in therapeutic pHSA solutions from four pharmaceutical companies using label-free quantification (Mikkat et al., 2020). Similarly, only 23 APs were detected in 28 batches of pHSA products from 27 domestic companies by ESI-Q-TOF MS (Ke et al., 2020). However, these studies had sample source limitations and did not include recombinant HSA (rHSA) samples. Despite the high purity (exceeding 99%) of rHSA produced by genetic recombination technology, the potential impact of trace amounts of APs cannot be ignored. Currently, there is limited knowledge about the clinical implications of APs in HSA solutions, and therefore, these proteins are generally considered safe for clinical use. However, there remains a possibility that these APs may affect the biological function of albumin in a potentially positive or negative manner. Current studies have not definitively established whether and how APs affect the therapeutic efficacy of HSA.

The four-dimensional label-free quantitative (4D-LFQ) proteomics method adds the separation of the collisional cross-section area drip (mobility) of the 4D peptide in addition to the three dimensions of retention time, mass to charge ratio, and ionic strength, thereby enabling higher quantitative accuracy, a wider dynamic range of proteome coverage and smaller sample mass per analyte (Meier et al., 2018; Prianichnikov et al., 2020). Given that the number of identifiable proteins in HSA preparations is constrained by the extreme dynamic range of protein concentrations, which can lead to suppression or masking of less abundant peptide ion signals by highly abundant albumin peptides, C18 Tip fractionation was applied before LC-MS/MS analysis in the present study to reduce the influence of albumin peptides. This proteomic method enables the detection of many proteins with low abundance (Li et al., 2021). Therefore, this method facilitates the rapid identification of a range of proteins and enables deeper insights using bioinformatic tools (Prianichnikov et al., 2020). In this study, we aimed to identify as many AP components in pHSA and rHSA products as we can using a 4D-LFQ proteomics technology and theoretically analyzed the functional effects of these AP components by bioinformatics methods.

Materials and Methods

HSA preparations

Two rHSA and six pHSA 20% preparations were donated by eight blood product companies in China: yeast-derived rHSA (YrHSA) from A company, oryza sativa derived rHSA (OsrHSA) from B company and pHSAs from C, D, E, F, G and H companies.

Sample preparation

Samples were removed from −80 °C, and an inhibitor (1% protease inhibitor) was added. The sample was sonicated and lysed. After centrifugation at 12,000 g and 4 °C for 10 min, the supernatant containing soluble proteins was carefully transferred to a fresh centrifuge tube. The molecular weight of proteins in each sample was analyzed by SDS-PAGE. According to the protein concentration measurement results, an equal amount of protein was taken for each sample and added to a centrifuge tube. 5 µL of 4× Loading buffer was added, then the total volume 20 µL was achieved by addition of 2% SDS. Loading: 1 µL pre-stained protein marker and 20 µL protein samples were loaded sequentially. The blank control was filled with 20 µL of 1× Loading buffer in the adjacent empty wells. Electrophoresis: the proteins were concentrated into a single line in the gel with15 mA/gel, approximately 15 min. Then, the separation gel was run at 35 mA until the dye reached the bottom of the gel. Staining and destaining: the gel was stained in Coomassie Brilliant Blue R-250 staining solution at room temperature for 2 h. Then, deatained by destaining solution and destained until the background became colorless and the bands became clear. Each sample was digested with same amounts, and then adjusted to the same volume with lysis solution (8 M urea). The samples were reduced using 5 mM dithiothreitol (30 min at 56 °C) and alkylated with 11 mM iodoacetamide (15 min at room temperature in darkness). After that, the urea concentration was adjusted to less than two M by adding 200 mM TEAB. Trypsin was added at 1:50 trypsin-to-protein mass ratio for an initial digestion overnight at 37 °C and 1:100 trypsin-to-protein mass ratio for a second 4 h digestion at 37 °C. At last, a Strata X SPE column was used to desalt the peptides.

C18 Tip fractionation

In order to separate the peptides into several fractions, a homemade tip column (C18, 3 µm, 150 Å, Agela) was used. First, was used to activate The tip column was activated by 50 µL 100% acetonitrile (ACN). Then, 50 µL 50% ACN and 50 µL phase A (98% H2O, 2% ACN, 0.1% formic acid) were used to equilibrate the tip column. The peptides were loaded into the tip twice in 50 µL phase A, and washed with phase A. Finally, the peptides were eluted into six fractions by increasing the ACN gradient: 7%, 10%, 17%, 23%, 28% and 50%. Then the fractions were combined into four fractions, Fra1 (7%, 28%), Fra2 (10%, 50%), Fra3 (17%) and Fra4 (23%), and dried by vacuum centrifugation.

LC-MS/MS analysis

After dissolved in solvent A, the tryptic peptides were directly loaded onto a homemade reversed-phase analytical column (100 µm i.d. × 25 cm) packed with 1.9 µm/120 ÅReproSil-PurC18 resins (Dr. Maisch GmbH, Ammerbuch, Germany). Solvent A (0.1% formic acid, 2% ACN in water) and solvent B (0.1% formic acid, 90% ACN in water) together made the mobile phase. The peptides were then separated with the following gradient: 0–68 min, 6%–23% B; 68–82 min, 23%–32% B; 82–86 min, 32%–80% B; 86–90 min, 80% B. An EASY-nLC 1,200 UPLC system from ThermoFisher Scientific was used with a constant flow rate of 500 nl/min. Then an Orbitrap Exploris 480 with a nano-electrospray ion source was used to analyze the separated peptides . And the FAIMS compensate voltage (CV) used was −45 V and −65 V. The Orbitrap detector was used to analyze precursors and fragments. The full MS scan resolution was set to 60,000 for a scan range of 400–1,200 m/z. 25 of the most abundant precursors were selected for further MS/MS analysis with 20 s dynamic exclusion, and the MS/MS scan was fixed as 110 m/z at a resolution of 15,000 with Turbot MT The HCD fragmentation was performed at a normalized collision energy of 27%. The automatic gain control target was set at 100%, with an intensity threshold of 50,000 ions/s and Auto maximum injection time.

Data analysis and bioinformatics

MaxQuant search engine (v.1.6.15.0) was used to process the resulting MS/MS data. Tandem mass spectra were searched against the UniProt database (http://www.uniprot.org) (Homo_sapiens_9606_SP_20230103.fasta, Oryza_sativa_subsp.japonica39947_PR_20221125. fasta and Saccharomyces_cerevisiae_4932_UP_20230316_seqkit.fasta) concatenating with the reverse decoy and contaminants database. The cleavage enzyme used was Trypsin/P, with two missing cleavages allowed. The minimum peptide length was set to seven, with a maximum of five modifications per peptide. The mass error tolerance was set as 20 ppm for precursor ions both in the first search and main search.

GO annotation was performed to annotate and analyze the identified APs with eggnog-mapper software (v2.0). The software is based on the EggNOG database (http://eggnog5.embl.de/#/app/home). The GO ID was extracted from the results of each protein annotation, and then the protein was classified according to cellular component, molecular function and biological process. The protein domains of the identified APs were annotated based on the Pfam database (https://www.ebi.ac.uk/interpro/entry/pfam/#table) and the corresponding PfamScan tool. WolF PSORT software was used to predict the subcellular localization of all identified APs. The NCBI COG (https://ftp.ncbi.nih.gov/pub/COG/) and EggNOG (http://eggnog5.embl.de/#/app/home) databases were used to provide a more comprehensive classification of species and protein sequences and to facilitate phylogenetic tree construction and functional annotation of each homologous gene cluster. The Kyoto Encyclopedia of Genes and Genomes (KEGG) database (http://www.kegg.jp/kegg/mapper.html) was used for KEGG pathway enrichment analysis. Fisher’s exact test was used to analyze the significance of KEGG pathway enrichment for the identified APs (using all proteins in the species database as the background), and P values <0.05 were considered significant. The Reactome database (https://reactome.org/) and the WiKipathways database (https://www.wikipathways.org/) were only used for the pathway enrichment analysis of the APs identified in the pHSA preparations.

Validation of proteomics results

A validation set (including the six pHSA samples) was detected by ELISA and according to the manufacturer’s protocols to verify the results of 4D label-free-based quantitative proteomics. The candidate APs validated by ELISA were haptoglobin (HP), hemopexin (HPX), serotransferrin (TF), transthyretin (TTR) and apolipoprotein M (ApoM). The selection of five proteins for ELISA validation was based on: (1) potential functional significance in the human body; (2) different functional categories or pathways; and (3) availability in commercial ELISA kits. The ELISA kits of HP, TF and TTR were purchased from Elabscience (Wuhan, China), and the HPX and ApoM ELISA kits were from FineTest (Wuhan, China).

Results

Molecular weight of proteins in HSA preparations

The SDS-PAGE analysis of proteins in HSA preparations revealed that the primary component, the albumin band was consistent with the expected molecular weight range of approximately 66.5 kDa (Fig. 1). Furthermore, the results demonstrated the presence of various other proteins with different molecular weights in all HSA preparations. Intriguingly, most APs exhibited molecular weights greater than albumin. Additionally, the pHSA preparations displayed a higher abundance and intensity of accompanying protein bands than the rHSA preparations.

Figure 1 Electrophoretic profile of proteins in HSA preparations using 12% SDS-PAGE under reducing conditions.

Lanes 1 and 2 are rHSA samples expressed in yeast (A) and O. sativa (B), respectively, and lanes 3–8 are pHSA samples from the six companies (C, D, E, F, G and H), respectively.

Identification of proteins in HSA preparations

The validation of MS data is shown in Figs. S1–S8. For all eight HSA samples, most of the peptides exhibited a distribution ranging from 7 to 20 amino acids, aligning with the general rule of fragmentation based on enzymatic hydrolysis and MS. The distribution of peptide lengths identified by MS met the quality control requirements. The coverage of most proteins was below 30%. In the shotgun (also called bottom-up) strategy, MS preferentially scans peptides that are more abundant. Consequently, there is a positive correlation between protein coverage and abundance in samples.

A total of 456 APs in all pHSA samples (company C: 186; D: 190; E: 166; F: 148; G: 342; and H: 182 Aps) were identified, whereas 52 APs in the YrHSA (yeast derived rHSA) sample and 152 in the OsrHSA were identified by 4D-LFQ proteomics (Tables S1–S8). The molecular weights of these APs identified in HSA preparations from three different sources are shown in Fig. 2. The frequency of 20–30, 10–20 and 50–70 kDa APs was most pronounced in the YrHSA, OsrHSA and pHSA samples after eliminating duplicates when merging data from the six pHSA samples, respectively.

Figure 2 Molecular weight distribution of APs in the HSA preparations from yeast, oryza sativa and human plasma.

The relative abundance and protein score of all APs in each sample were calculated (Table S9). According to the relative abundance value of each AP, three proteins (84.6% K7_Dit1p, 8.7% DASH complex subunit ASK1, 5.1% Nucleolar protein 9) in YrHSA and eight proteins (63.3% Cytochrome c oxidase subunit 5C, 11.7% LOC_Os10g39430, 8.0% Terpene cyclase, 2.3% Os11g0303200, 1.9% Superoxide dismutase (Cu-Zn) chloroplastic, 1.6% Superoxide dismutase (Cu-Zn) 1: SODCC1, 1.1% Lactoylglutathione lyase and 1.0% Early nodulin-like protein 1) in OsrHSA had a relative abundance value greater than 1% (Fig. 3). Among the six pHSA samples, the eight proteins with the highest relative abundance in each pHSA sample included HP, HPX, TF, TTR, plasma protease C1 inhibitor (SERPING1), alpha-1-acid glycoprotein 1 (ORM1), keratin type I cytoskeletal 9 (KRT9), keratin type II cytoskeletal 1 (KRT1), hemicentin-1 (HMCN1), beta-2-glycoprotein 1 (APOH), apolipoprotein A-II (APOA2), alpha-2-HS-glycoprotein (AHSG), afamin (AFM) and alpha-1B-glycoprotein (A1BG). Notably, the relative abundance of HP, HPX and TTR in all pHSA samples exhibited relatively high levels (Fig. 4).

Figure 3 Percentage relative abundance of the three and eight most abundant APs in rHSA samples from yeast (A) and O. sativa (B).

Other APs with a relative abundance of less than 1% are also shown.

Figure 4 Percentage relative abundance of the eight most abundant APs in each pHSA sample from the six companies (C, D, E, F, G and H).

Other APs at relatively much lower abundance are not shown.

All identified APs in the pHSA samples were analyzed by UpSetR (Fig. 5). Ninety-six APs were identified in the six pHSA samples. Additionally, 156 proteins were identified exclusively in the pHSA sample from G company. Furthermore, 22, 19, 15, 12 and 11 proteins were identified exclusively in the pHSA samples from D, H, C, E and F companies, respectively.

Figure 5 An UpSetR plot of variants across the six pHSA samples (companies C, D, E, F, G and H).

The size of the intersections is visually represented using a bar chart placed on top of the matrix. Each column in the matrix aligns with a specific bar in the chart. Additionally, a separate bar chart is displayed to the left of the matrix, illustrating the size of each group. In this representation, each row represents a distinct group, and the columns represent intersections in the matrix. A black-filled or light grey circle is placed in the corresponding matrix cell to indicate the presence or absence of a group within an intersection. The black-filled circle signifies that the group is part of the intersection, whereas the light grey circle indicates its exclusion. A vertical black line connects the topmost and bottom-most black circles within each column to highlight the relationships based on columns. This marking emphasizes the column-based relationships between the groups and their intersections. The data associated with each column in the matrix corresponds to the number of expressed proteins in the groups represented by the black circles. This information can be used to select proteins within specific intersections for further analysis or characterization.

Bioinformatics analysis of APs in HSA

To comprehensively understand the functional properties of the identified APs (merging data from the six pHSA samples and removing duplicates) in pHSA, various aspects of these proteins were annotated, including Gene Ontology (GO), protein domain, KEGG pathway, COG/KOG functional classification, subcellular localization, Reactome and WikiPathways (Table S10). Enrichment analysis of the identified APs was then performed (note: bioinformatics analysis of APs identified in rHSA preparations are available to examine in Files S1 and S2).

According to Gene Ontology, the APs in pHSA clustered primarily into 43 GO functional categories, including 17 biological processes, 12 cellular components and 14 molecular functions (Fig. S9). Subcellular localization analysis showed that these proteins in pHSA samples mainly included 132 extracellular proteins, 111 cytoplasm-related proteins, 92 nucleus-related proteins, 39 plasma membrane-related proteins, 32 mitochondria-related proteins, 18 cytoplasm nucleus-related proteins, 18 endoplasmic reticulum-related proteins and 14 other undefined proteins (Fig. S10). Functional classification of these APs was carried out using the COG/KOG protein databases. The results revealed that the APs in pHSA were classified into 24 COG/KOG categories, among which “posttranslational modification, protein turnover, chaperones” represented the largest group and involved 51 proteins (Fig. S11). KEGG pathway analysis showed that these proteins were involved in 42 pathways: four cellular processes, three environmental information processing, three genetic information processing, 11 human diseases, 11 metabolism and 11 organismal systems (Fig. S12).

GO enrichment analysis revealed that the most significantly enriched biological processes were the regulation of proteolysis, negative regulation of endopeptidase activity and negative regulation of peptidase activity. The top two cellular components were enriched in the extracellular region and extracellular space. The main molecular functions were endopeptidase inhibitor activity, peptidase inhibitor activity and endopeptidase regulator activity (Fig. 6 and Table S11).

Figure 6 Biological process (A), cellular component (B) and molecular function (C), as determined by the GO term enrichment analysis of the APs identified in pHSA.

Enrichment analysis of the protein domains in the APs revealed that the most enriched term was serpin (serine protease inhibitor) (Fig. 7A and Table S12). Most of the significantly enriched domains are directly related to the regulation of serine and other proteases that play roles in coagulation, fibrinolysis, inflammation, wound healing and tissue repair. The enrichment analysis of the KEGG pathway (Fig. 7B and Table S13) revealed that the top four enriched pathways were complement and coagulation cascades, Staphylococcus aureus infection, cholesterol metabolism and the estrogen signaling pathway. Moreover, 19, 19, 10 and 16 APs, respectively, were identified to be involved in the aforementioned pathways. The pathway enrichment results showed that these APs were involved in three primary Reactome pathways: neutrophil degranulation, innate immune system and immune system (Fig. 7C and Table S14). Furthermore, enrichment analysis revealed that the most significantly enriched WikiPathway was complement and coagulation cascades, and the next enriched pathways were the network map of the sarscov2 signaling pathway and aerobic glycolysis (Fig. 7D and Table S15).

Figure 7 Enrichment analysis of protein domains (A), KEGG pathway (B), Reactome pathway (C) and WikiPathway (D) of APs identified in pHSA.

Protein–protein interaction network analysis of APs and albumin in pHSA

We constructed a PPI proteomic network using the STRING database to understand better the interactions among the APs in pHSA and albumin. PPI analysis revealed that 49 APs interacted with albumin, of which 24 might be upstream interacting proteins and 25might be downstream interacting proteins (Tables S16 and S17). The robust and crosstalk signaling cascades are mapped in Fig. 8. Additionally, two closely interacting protein clusters were screened by the MCODE method. KEGG pathway enrichment analysis was performed on these two protein clusters, and the top two significantly enriched pathways was the biosynthesis of amino acids and cholesterol metabolism.

Figure 8 Protein–protein interaction network analysis of the APs.

With a confidence score > 0.7, 312 APs (merging data from the six pHSA samples and remov ing duplicates) were included in the analysis. The STRING database was used to annotate the functional interactions of the identified APs.

Validation of proteomics results

ELISA detection was performed for five proteins, i.e., HP, HPX, TF, TTR and ApoM, to validate the quantitative data obtained by 4D label-free quantitative proteomics. The ELISA results revealed that HP, HPX, TF and TTR were detectable in pHSAs but not in rHSAs, which is consistent with the results of 4D label-free quantitative proteomics (Fig. 9). However, because of the extremely low levels of ApoM and TF, it was challenging to reach the detection limit of the ELISA assay kit. TF was not detected in the pHSA sample from G company (Fig. 9D), and ApoM was not detected in all pHSA samples by ELISA. Furthermore, the ELISA results indicated significant differences in HP, HPX, TF and TTR levels among pHSA samples obtained from different manufacturers.

Figure 9 Verification of APs identified in HSA samples (rHSA samples from A and B companies, pHSA samples from C, D, E, F, G and H companies) by ELISA.

(A) HP, (B) HPX, (C) TTR and (D) TF. Data represent the mean ±  SD for six pHSA samples and two rHSA samples analyzed using five replicates.

Discussion

In this study, APs in pHSA and rHSA preparations were identified and analyzed by 4D label-free quantitative proteomic technology. Similar to previous reports (Gay et al., 2010; Ke et al., 2020; Mikkat et al., 2020), many APs were identified in the HSA samples. However, in contrast to previous studies, HSA preparations were derived from more sources (particularly rHSA samples) and a larger set of APs were identified. Our results revealed that pHSA samples contained more types of APs compared with rHSA samples, with the YrHSA sample exhibiting the lowest number of APs. This result is consistent with the high purity standard of YrHSA (>99% according to the manufacturer’s quality inspection report). The purity standard of OsrHSA (according to the manufacturer’s quality inspection report) and pHSA (according to the Chinese Pharmacopoeia) was >96%, resulting in relatively more types of APs detected in OsrHSA and pHSA than in YrHSA. The different levels and types of APs in HSA preparations may be attributed to various factors, such as plasma source (for pHSA), production and purification processes.

The top three APs (K7_DIT1, AWRI1631_111740, and NOP9) with the highest relative abundance identified in YrHSA are nuclear-associated proteins, whereas the first three APs (COX5C, LOC_Os10g39430 and Os08g0223900) identified in OsrHSA are cytoplasm-associated proteins. These APs identified in rHSA are host cell proteins (HCPs). HCPs are critical factors that affect the quality of recombinant protein drugs (Goey, Alhuthali & Kontoravdi, 2018). Sandberg et al. (2006) found that some HCPs with catalytic activity potentially disrupt the structure of drugs by enzymatic hydrolysis or modification of protein drugs, leading to the inactivation of the protein drug. Moreover, HCPs were found to facilitate multimer formation of protein drugs (Eon-Duval, Broly & Gleixner, 2012), and excessive HCPs have been linked to adverse reactions in various clinical trials (Gutiérrez, Moise & De Groot, 2012). Although the content of APs in rHSA is very low, the potential adverse effects of HCPs in clinical use cannot be ignored. Therefore, more sensitive and accurate methods for identifying and quantifying HCPs are necessary to ensure the safety and efficacy of recombinant protein drugs.

The top eight APs with the highest relative content in every pHSA are shown in Fig. 4, and a total of 14 proteins were included. Except for HMCN1, which was exclusively identified in the pHSA from F company, the remaining 13 APs were detected in the six pHSAs (Tables S1–S8). HP, HPX and TTR were consistently among the top eight APs in each pHSA, which is similar to previous studies examining APs in pHSAs (Gay et al., 2010; Ke et al., 2020; Mikkat et al., 2020). The APs in pHSA are derived from human plasma and are generally considered safe in clinical applications because they typically do not cause adverse reactions. Hence, in this study, we mainly focused on the biological functions of the APs in pHSAs. Among the APs, HP was identified as the top AP with the highest relative abundance in all pHSA preparations. HP captures and binds free plasma hemoglobin to facilitate the hepatic recycling of heme iron and to prevent kidney damage (Ratanasopa et al., 2013). HP also acts as an antioxidant, possesses antibacterial activity and plays a role in modulating the acute phase response (Di Masi et al., 2020). Hemoglobin/HP complexes can be rapidly cleared by the CD163 receptor on liver Kupfer cells via endocytic lysosomal degradation (Graversen, Madsen & Moestrup, 2002). HPX is a plasma glycoprotein that scavenges heme. This protein is important in hemolytic disorders and under hemorrhagic conditions because of the increase in the release of hemoglobin, which elevates labile heme levels and leads to oxidative stress (Lechuga et al., 2022). Consequently, HPX has potential applications in both therapy and diagnostics. TTR binds and transports thyroid and steroid hormones to maintain their stability and biological activity (Schussler, 2000). Additionally, the retinol transport function of TTR is essential for normal visual system development. TTR also promotes nerve cell survival, inhibits neurodegenerative disease progression and protects against oxidative stress through vitamin A and antioxidant transport (Buxbaum et al., 2008; Magalhães, Eira & Liz, 2021). From a biological perspective, the presence of the aforementioned three APs in pHSA preparations is important for the health of patients receiving pHSA treatment. Furthermore, the analysis of PPI revealed that ALB, HPX, TTR and another 15 APs formed a closely interconnected protein cluster. The KEGG pathway enrichment analysis highlighted that “biosynthesis of amino acids” was the most significantly enriched pathway within the protein clusters identified.

KEGG pathway classification analysis showed that the second-level category cell growth and death in cellular processes involved 26 APs identified in pHSA preparations (Fig. S12 and Table S18). Among them, the following 11 APs are related to cell growth: Cathepsin C, Calmodulin-like protein 3, tyrosine 3-monooxygenase, calmodulin-like protein 5, Heme Oxygenase 1, Calpain1, TF, Heat Shock Protein 90 Alpha Family Class B Member 1, glutamine synthetase, cathepsin D and cyclin B1. This observation corroborates a recent study showing that plasma-derived pHSA exhibits more favorable effects on stem cell cultivation compared with rHSA (Chen & Xu, 2021). This distinction may be because these aforementioned APs are related to mammalian eukaryotic cell growth, which were not present in rHSA. This report also found significant differences in the activity of pHSA from different manufacturers to promote stem cell expansion (Chen & Xu, 2021). Additionally, this variability may be attributed to differences in APs present in pHSA from different manufacturers. According to the functional classification results of COG/KOG, 10 APs (ABCA13, Lysophospholipase 1, TTR, lecithin-cholesterolacyltransferase, Diazepam binding inhibitor, Phospholipase A2 Group IVB, Acid phosphatase 3, Phospholipase C Gamma 2, Fatty acid-binding protein 5, mevalonate kinase) (Fig. S11 and Table S19) are involved in lipid transport and metabolism. These APs may regulate the transport, metabolism and biochemical processes of lipids within cells. Moreover, these proteins potentially exhibit functions such as binding, transport, catalysis and regulation of lipid molecules to maintain normal lipid metabolism and cellular function. Previous studies conducted by Keenan et al. (1997) and Schiller et al. (2008) found that biologically active lipids in pHSA promote the growth of normal rat kidney (NRK) cells and the cellular survival of peripheral blood mononuclear cells (PBMCs). These APs involved in lipid transport and metabolism may regulate the transport, metabolism and biochemical processes of intracellular lipids. Therefore, these APs in HSA preparations may have specific effects on the in vitro culture of certain cells. These APs potentially influence lipid metabolism and cellular functions in vitro, affecting cell proliferation, viability and differentiation. Further research into the functions and mechanisms of these APs will contribute to a deeper understanding of the regulatory mechanisms of lipid metabolism in cell cultures and guide the improvement of in vitro cultivation techniques.

We have also identified ApoM in pHSA samples from C and H using 4D label-free quantitative proteomic technology (Tables S3 and S8). ApoM primarily binds to high-density lipoproteins (HDLs) and specifically retains sphingosine-1-phosphate (S1P) within its hydrophobic pocket (Christoffersen et al., 2011). A recent study (Ruiz, Okada & Dahlbäck, 2017) indicated that ApoM-bound S1P plays a crucial role in the anti-apoptotic activity of HDL and enhances endothelial function. Mechanistically, the cooperation between S1P1 and S1P3 was required for HDL/ApoM/S1P complex-mediated anti-apoptosis. Additionally, the phosphorylation of AKT and ERK is necessary to achieve the anti-apoptotic effects of the HDL/ApoM/S1P complex. ApoM-bound S1P is a key component of HDL and is responsible for several HDL-associated protective functions in the endothelium, such as regulating adhesion molecule abundance, leukocyte-endothelial adhesion and the endothelial barrier (Ruiz et al., 2017). Noteworthy, approximately 50% to 60% of S1P in the plasma binds to ApoM, whereas approximately 30% to 35% binds to serum albumin (Murata et al., 2000; Winkler et al., 2019). Therefore, the potential significance of ApoM in pHSAs deserves attention and further research.

Conclusion

In summary, we initially identified 456 different APs from six pHSA samples using a 4D label-free quantitative proteomic technique. Bioinformatics analysis (e.g., GO, KEGG, PPI) revealed these APs were associated with various physiological processes, including binding free hemoglobin, exhibiting antioxidative and antibacterial properties, regulating cell growth and death and facilitating lipid transport and metabolism. From a biological functional perspective, these plasma-derived APs may benefit human health in certain aspects (e.g., antioxidant, antibacterial). Importantly, these APs were absent in rHSA preparations. Consequently, in clinical therapeutic applications, pHSA may offer potential advantages in efficacy and safety compared with rHSA. Moreover, despite low levels, the adverse reactions caused by residual host cell proteins in rHSA should not be underestimated. 4D Label-free quantitative proteomics provides an efficient and sensitive method for detecting low-abundance APs in HSA preparations, enhancing our understanding of their presence and potential effects.

Supplemental Information

Supplemental Information 1 Figure 2 raw data

Supplemental Information 2 Supplemental materials

Supplemental Figures:

Figure S1A: Peptide length, peptides per protein, distribution of coverage (%) and MW (kDa) of the LC-MS/MS analysis of rHSA from company A.

Figure S2B: Peptide length, peptides per protein, distribution of coverage (%) and MW(kDa) of the LC-MS/MS analysis of rHSA from company B.

Figure S3C: Peptide length, peptides per protein, distribution of coverage (%) and MW(kDa) of the LC-MS/MS analysis of pHSA from company C.

Figure S4D: Peptide length, peptides per protein, distribution of coverage (%) and MW(kDa) of the LC-MS/MS analysis of pHSA from company D.

Figure S5E: Peptide length, peptides per protein, distribution of coverage (%) and MW(kDa) of the LC-MS/MS analysis of pHSA from company E.

Figure S6F: Peptide length, peptides per protein, distribution of coverage (%) and MW(kDa) of the LC-MS/MS analysis of pHSA from company F.

Figure S7G: Peptide length, peptides per protein, distribution of coverage (%) and MW(kDa) of the LC-MS/MS analysis of pHSA from company G.

Figure S8H: Peptide length, peptides per protein, distribution of coverage (%) and MW(kDa) of the LC-MS/MS analysis of pHSA from company H.

Figure S9: GO enrichment analysis of the APs in pHSA.

Figure S10: Subcellular localization prediction of the APs in pHSA.

Figure S11: COG/KOG enrichment analysis of the APs in pHSA.

Figure S12: KEGG pathway enrichment analysis of the APs in pHSA.

Supplemental Tables:

Table S1A: The protein and peptide identified in rHSA from company A.

Table S2B: The protein and peptide identified in rHSA from company B.

Table S3C: The protein and peptide identified in pHSA from company C.

Table S4D: The protein and peptide identified in pHSA from company D.

Table S5E: The protein and peptide identified in pHSA from company E.

Table S6F: The protein and peptide identified in pHSA from company F.

Table S7G: The protein and peptide identified in pHSA from company G.

Table S8H: The protein and peptide identified in pHSA from company H.

Table S9: The relative abundance and protein score of all APs in each sample.

Table S10: Protein annotation of the identified APs in pHSA.

Table S11: GO enrichment of the identified APs in pHSA.

Table S12: Protein domain enrichment of the identified APs in pHSA.

Table S13: KEEG enrichment of the identified APs in pHSA.

Table S14: Reactome pathways analysis of the identified APs in pHSA.

Table S15: WikiPathway analysis of the identified APs in pHSA.

Table S16: The nodes of PPI proteomic network analysis with STRING database of the identified APs in pHSA.

Table S17: The links of PPI proteomic network analysis with STRING database of the identified APs in pHSA.

Table S18: KEEG classification of the identified APs in pHSA.

Table S19: COG classification of the identified APs in pHSA.

Supplemental Information 3 Figure 9 raw data

Supplemental Information 4 Uncropped gel

Supplemental Information 5 Description of supplementary data

Abbreviations

HSA human serum albumin

APs accompanying proteins

pHSA plasma-derived human serum albumin

rHSA recombinant human serum albumin

YrHSA yeast-derived rHSA

OsrHSA oryza sativa derived rHSA

MS mass spectrometry

4D-LFQ four-dimensional label-free quantitative

ACN acetonitrile

HP haptoglobin

HPX hemopexin

TTR transthyretin

TF serotransferrin

ApoM apolipoprotein M

GO Gene Ontology

COG Clusters of Orthologous Groups

KOG Karyotic Orthologous Groups

KEGG Kyoto Encyclopedia of Genes and Genomes

PPI protein–protein interactions

ELISA enzyme-linked immunosorbent assay

4D-LFQ 4D label-free quantitative

HCPs host cell proteins

HDLs high-density lipoproteins

S1P sphingosine-1-phosphate

Additional Information and Declarations

Competing Interests

Author Contributions

Data Availability

Jun Xu and Lu Cheng are employed by Shanghai RAAS Blood Products Co., Ltd. The authors declare there are no competing interests.

Li Ma conceived and designed the experiments, performed the experiments, analyzed the data, prepared figures and/or tables, authored or reviewed drafts of the article, and approved the final draft.

Peng Jiang performed the experiments, analyzed the data, prepared figures and/or tables, and approved the final draft.

Zongkui Wang analyzed the data, prepared figures and/or tables, and approved the final draft.

Qing Liu performed the experiments, prepared figures and/or tables, and approved the final draft.

Jun Xu performed the experiments, prepared figures and/or tables, and approved the final draft.

Lu Cheng performed the experiments, prepared figures and/or tables, and approved the final draft.

Pan Sun conceived and designed the experiments, authored or reviewed drafts of the article, and approved the final draft.

Xi Du conceived and designed the experiments, authored or reviewed drafts of the article, and approved the final draft.

Changqing Li conceived and designed the experiments, authored or reviewed drafts of the article, and approved the final draft.

The following information was supplied regarding data availability:

The MS proteomics data is available at ProteomeXchange Consortium via the PRIDE partner repository with the dataset identifier PXD045339.

Raw data is available in the Supplemental Files.

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
