# Peer review of "Protein composition analysis of human plasma-derived and recombinant human serum albumin preparations based on 4D label-free proteomics"

_PeerJ, doi:10.7717/peerj.19624_

## Round 0.1 · original submission · Major Revisions

Please address the concerns of both reviewers and amend the manuscript accordingly.

Reviewer 1 ·

Basic reporting

The language of the paper is generally correct and understandable.

The extensive section on mass spectrometry (lines 82-103) in the introduction of the paper could be greatly abbreviated to information on the application of MS for similar studies. Literature relating to methodology (fractionation, MS measurements) should be included in the description of methods.

In the supplementary data, there are no descriptions of the tables, figures and files contained therein. Brief references to this information in the text of the paper, are not sufficient.

Some of the references are incomplete (Lechuga, G. C et al. Lines 475-476 or Li, J et al. Lines 477-479) these are articles in electronic versions of journals so article numbers should be given (there are no page numbers).

The raw data have been deposited in the public repository.

Experimental design

The authors' motivation is not entirely clear; what is the purpose of this research? Of course, it is valuable that the authors have examined preparations containing recombinant proteins. However, there is no clear purpose. It would be understandable for the reviewer if the authors focused on identifying as many contaminating proteins as possible. In this case, the authors should consider enriching the sample by removing albumin (e.g. with a column containing the appropriate antibodies) so that proteins present in low concentrations can be identified. The fractionation used by the authors is a very simple but also one of the basic/routine methods. Successful identification would allow for more effective and reliable protein quantification.
From the brief description of the samples tested in Section 2.1, it is not clear whether the products meet the requirements of the previously cited Pharmacopoeias. If the samples meet different purity criteria, comparing them is of questionable quality. It is rather strange and unacceptable that the authors do not mention the names of the manufacturers of the samples examined. The description of the samples should be complete and refer to the quality declared by the manufacturer, as well as who the manufacturer is. Some information on this subject is provided in chapter 4 (lines 304-308), but it is very fragmentary and concerns only some samples. Furthermore, the description “Two rHSA and six pHSA 20% preparations” (line 106) suggests that the authors examined 20% albumin solutions. There is no information about the solution in which the albumin was dissolved, what this solution contained and how any components of this solution could have influenced the analyses.
According to the description on line 110, a protease inhibitor was added to the albumin solution tested. This will inevitably affect the trypsin used in the subsequent steps. Why was this inhibitor added? If products registered as medicines are being tested, the manufacturer should ensure that the product is stable (this refers to the previous question regarding the composition of the sample).
According to the current recommended standards criteria for proteomic studies (J Proteome Res. 2019 Oct 21;18(12):4108-4116. doi: 10.1021/acs.jproteome.9b00542) the minimum requirements for reliable identification of proteins are the identification of at least two peptides of a minimum length of 9 (or one peptide 18 residue or more long) amino acid residues each, and a value of the False Discovery Rate (FDR) parameter less than 1.0.
The vast majority of proteins were identified based on only one peptide (Fig. S1-8A-F) and the vast majority of these peptides are not 18 residues or longer (Tables S1-8A-F). As described in line 167, the FDR parameter had a value below 1.0. In summary, most of the protein identifications presented in the paper do not meet the requirements described in the recommendations cited above, so the quality of these identifications is questionable.
The authors perform a quantitative analysis using the 4D label-free method. Firstly, it is necessary to perform a minimum of three repetitions of each measurement in quantitative studies so that at least the range of error resulting from the measurement can be determined. Unfortunately, in the described studies, the authors only performed single tests. The description of the methods does not contain information about the repetitions used, and the presented results do not contain information about the standard deviation of the values.
The 4D label-free method is a method of relative concentration determination, where the change in concentration is described in relation to a certain reference point (e.g. in relation to the concentration of a specific protein). The description of the methodology does not contain information about what this reference point was.

Validity of the findings

The authors performed concentration tests for four proteins using ELISA. However, in many cases there are gross discrepancies between the results of MS measurements (Figure 4) and measurements using the ELISA test (Figure 9). For example, for the TF protein for sample F, the concentration in the ELISA test was the highest of all the analyzed proteins (see Figure 9D), but from the MS measurements, the relative concentration (see note above) was one of the lowest. The discrepancy in results between the methods is not fully addressed in the discussion or conclusions.
A significant part of the results and discussions concerns the functionality of the identified proteins. It is not entirely clear what the purpose of these very extensive analyses is. The only rational purpose of such analyses is to identify contaminating proteins that would affect the product's activity. However, these are standard analyses based on GO classification based on KEGG and completely unrelated to the impact of contaminants on the biological activity of the preparation.
As described above, the identification of many of the proteins described is of questionable quality (see comment on the reliability of identification J Proteome Res. 2019 Oct 21;18(12):4108-4116).

Additional comments

None

·

Basic reporting

The authors present a detailed analysis of accompanying proteins (APs) from several sources of albumin purification including via manufacturer purification from human plasma and through recombinant means from yeast and rice.

There are several typos throughout the text. In the abstract, "M ost" should be Most and "v alidation" should be validation. On page 5, line 151, "Turbot MT" should be turbo TMT.

Page 4, line 92: The explanation of 4D-LFQ proteomics could use additional description. This term is often used in regard to timsTOF platforms and on instruments in which a unique measurement can be attributed to the analytes collisional cross section which drives a measurable separation between analytes. As the authors are using high-field asymmetric waveform ion mobility spectrometry (FAIMS) which can separate analytes by m/z and z but does not provide a unique fourth measurement, the authors should be clear to differentiate their data acquisition strategy. At a minimum, the authors need to explicitly state the four dimensions of separation (chromatography (RT), ion mobility, m/z, and intensity).
In addition to the papers cited in this sentence, the authors should consider citing relevant FAIMS literature, such as:
DOI: 10.1074/mcp.TIR118.000862
DOI: 10.1021/acs.analchem.8b02233
DOI: 10.1021/acs.analchem.8b05399

Page 4, line 93: "smaller sample volumes" should be smaller sample mass, as the required peptide loading mass to achieve a certain quantitation threshold is what typically dictates a methods sensitivity. Sample volume is not relevant since the concentration can be adjusted.

Page 7, paragraph 3: The authors should be clear that they have filtered out albumin signal in their relative abundance percentages. Further the current sentence structure in listing out the top APs is a bit complicated. Perhaps the authors can leave out some information and only write out the gene name OR protein description.

Page 8, line 251-257: The authors should include some description as to why PTM, protein turnover, and chaperones represented the largest group in the COG/KOG categories. Are kinases, proteases, or other PTM-related enzymes featured in their identifications?

This reviewer was unable to access the mass spectrometry RAW data using the PRIDE credentials provided.

Experimental design

Previous studies of albumin preparations have focused on albumin yield, purity, oxidation status, and post-translational modification. The authors here focus on heightened analytical sensitivity and bioinformatic techniques to identify APs which may affect therapeutic efficacy of albumin preparations. The research is well defined, relevant, and fills an identified knowledge gap.

Validity of the findings

The manuscript does not explicitly clarify whether the identified APs are co-purified due to their association with HSA (e.g., forming stable complexes) or if they are merely among the most abundant proteins in the original plasma, rice, or yeast extracts. This distinction is crucial, as some APs (e.g., IgGs in plasma) might be present due to sheer abundance rather than direct interactions with albumin. It appears like the second most abundant proteins in the Figure 1 gel may be IgGs. Additional experimental evidence, such as co-immunoprecipitation or crosslinking studies, could help clarify this point.

The authors use STRING database analysis to cluster APs and identify 49 APs interacting with albumin. However, it is unclear whether these represent true protein-protein interactions (PPIs) or predicted associations based on network analysis. STRING incorporates both experimental and predicted interactions, including co-expression and text-mining data, which do not necessarily indicate direct binding. Page 8, line 280-282: The authors should clarify what they mean by "24 were upstream", and "25 were downstream interacting proteins".

Furthermore, the study does not provide experimental validation of these interactions beyond ELISA, which only confirms the presence of specific APs in the samples. Without additional evidence, such as co-immunoprecipitation, affinity purification-mass spectrometry (AP-MS), or structural modeling, the claim that these proteins interact directly with albumin remains unsubstantiated. To strengthen their conclusions, the authors should clarify whether they are proposing functional associations rather than direct PPIs and consider validating key interactions experimentally.

One helpful addition to the overall story, either in replacement or addition of Figure 3, would be a "waterfall" plot ranking all quantified protein groups by log10(intensity). An even better approach would be to plot the measured AP proteins' abundances against known plasma protein concentrations. This would give the reader an appreciation for the dynamic range and a sense of whether the APs identified and quantified are simply among the most abundant proteins in plasma, or if they are specifically enriched in the preparation due to a complex formation or aggregation with albumin.

---

## Round 0.2 · accepted · Accept

In my view, all the issues pointed out by the reviewers were addressed, and the manuscript was revised accordingly. Therefore, the revised manuscript is acceptable now.